# Preoperative Radio(Chemo)Therapy in Breast Cancer: Time to Switch the Perspective?

**Angel Montero** \*,† **and Raquel Ciérvide** † 

Department of Radiation Oncolo Gy, HM Sanchinarro University Hospital, HM Hospitales, 28050 Madrid, Spain
\* Correspondence: angel.monteroluis@gmail.com or amontero@hmhospitales.com
† These authors contributed equally to this work.

**Abstract:** Aim: Radiation therapy represents, together with surgery and systemic treatment, the triad on which the current management of patients with breast cancer is based, achieving high control and survival rates. In recent years we have witnessed a (r)evolution in the conception of breast cancer treatment. The classic scheme of surgery followed by systemic treatment and radiotherapy is being subverted and it is becoming more and more frequent to propose the primary administration of systemic treatment before surgery, seeking to maximize its effect and favoring not only the performance of more conservative surgeries but also, in selected cases, increasing the rates of disease-free survival and overall survival. Radiotherapy is also evolving toward a change in perspective: considering preoperative primary administration of radiotherapy may be useful in selected groups. Advances in radiobiological knowledge, together with technological improvements that are constantly being incorporated into clinical practice, support the administration of increasingly reliable, precise, and effective radiotherapy, as well as its safe combination with antitumor drugs or immunotherapy in the primary preoperative context. In this paper, we present a narrative review of the usefulness of preoperative radiotherapy for breast cancer patients and the possibilities for its combination with other therapies.

**Keywords:** breast cancer; preoperative radiotherapy; neoadjuvant radiochemotherapy



## 1. Introduction

Breast cancer is the most frequently diagnosed neoplasm with 530,000 new cases in 40 European countries (for each of the 39 UN-defined European countries and Cyprus) and the leading cause of female cancer mortality in most countries. There were considerable variations in the estimated incidence rates of breast cancer among European countries (from 71 to 194 per 100,000) in 2020 [1].

Surgery, radiation, and systemic treatment remain the cornerstones for achieving locoregional control and breast cancer survival. In recent years, different ways of combining these three strategies have been developed, altering the sequence of their administration and considering the characteristics of each case in order to increasingly personalize and tailor treatments.

## 2. Neoadjuvant Systemic Treatment

Neoadjuvant systemic treatment or primary systemic treatment (PST) for breast cancer was primarily conceived to allow more conservative surgeries for those tumors initially considered to be unresectable. However, its use was later extended to evaluate tumor shrinkage and to rapidly assess pathological and clinical response, which are also associated with both progression-free and overall survival [2].

This strate Gy is of special interest in Her-2 enriched (HER2+) and triple-negative (TN) breast cancer patient subgroups, who are the best responders to neoadjuvant chemotherapy as compared to those with luminal subtypes. Different studies have shown that achieving

a pathological complete response (pCR) after PST, specifically in the subgroup of patients with TN or HER2-positive tumors, is associated with significant survival gains [3–5].

At least three phase III studies (MD Anderson Cancer Center neoadjuvant trastuzumab trial, Neoadjuvant Herceptin (NOAH) trial, Gepar-Quattro trial) compared neoadjuvant chemotherapy alone to the same chemotherapy plus trastuzumab and showed a significant increase (65%) in pathological complete response [6–10].

In addition, large randomized clinicaltrials have demonstrated that dual HER2-targeted blockade with trastuzumab/lapatinib [9] and trastuzumab/pertuzumab [10] works synergistically, enhancing final response. The Neosphere study [11], which analyzed the combination of trastuzumab and pertuzumab plus chemotherapy in h patients, showed that patients who achieved pathological complete response had longer progression-free survival compared to patients who did not (85% vs. 76%; hazard ratio 0.54 (95% CI 0.29–1.00)). The significant increase in pCR and the impact on survival rates positioned this combination as the new standard. The Tryphaena study [12] assessed the differences between distinct chemotherapy schemes when combined with double Her-2 blockade, achieving pCR in 57–66% of patients without observing any variations directly attributable to the chemotherapy schedule [13]. The Berenice study [11], which combined neoadjuvant pertuzumab, trastuzumab, and anthracycline-taxane-based chemotherapy, achieved the same abovementioned pCR rates, confirming, once again, safety in terms of cardiac tolerance.

Likewise, patients with TN breast cancer have a worse prognosis associated with lower overall survival. Neoadjuvant systemic treatment has been proposed as an attractive alternative in these patients due to the known relationship between pCR rates and overall and progression-free survival. Thus, pCR is considered to be a surrogate survival marker for TN breast tumors. The use of conventional chemotherapy regimens as part of the PST with the combination of adriamycin, paclitaxel, and cyclophosphamide has been associated with pCR rates of 35–45% [12]. The addition of platinum salts to PST schemes showed an increase in these rates. A recent meta-analysis of nine randomized studies including more than 2000 women with TN breast cancer reported that the addition of platinum compounds to the PST increased the pCR rate from 37% to 52.1% ($p < 0.001$) [14].

In addition, the KeyNote-522, Impassion 031, and GeparNUEVO studies recently demonstrated that the addition of immunotherapy (pembrolizumab, atezolizumab, or durvalumab, respectively) plus chemotherapy achieved a significant increase in the percentage of patients with a pathological complete response compared to those who received placebo-chemotherapy [15–17], although the NeoTRIP Michelangelo trial failed to show any benefit from the addition of atezolizumab to nab-pacliatxel in TNBC [18]. Finally, the I-SPY-2 trial (Investigation of Serial Studies to Predict Your Therapeutic Response with Imaging and Molecular Analysis 2, NCT01042379) represents a new generation of clinical trials pursuing personalized medicine for breast cancer patients by evaluating response rates to multiple new agents based on imaging characteristics and tumor biomarkers [19].

### 2.1. Neoadjuvant Radiotherapy

About eight out of ten patients with breast cancer are treated with ionizing radiation at some point. Advances and improvements in locoregional treatments of breast cancer, surgery, and radiotherapy have contributed decisively to decreasing locoregional recurrences and distant recurrences while increasing breast cancer and overall survival. The results of the Early Breast Cancer Trialists' Collaborative Group (EBCTCG) meta-analysis, conducted on 17 studies that included 10,801 women, showed that radiotherapy significantly decreased the risk of recurrence, locoregional or distant, at 10 years, and breast cancer-specific mortality at 15 years [20]. In 2014, the same group published an update of results with a greater follow-up of over 8135 women; the results showed a significant reduction in the likelihood of locoregional and/or distant recurrence in those women with tumor lymph node involvement who were irradiated [21]. These benefits were observed in all groups of patients, both in patients with one to three affected lymph nodes and in those with metastases in more than four lymph nodes. The benefits observed at 10 years

resulted in a significant increase in breast cancer survival at 20 years and were independent of the administration or not of systemic treatment. According to the authors, "one death from breast cancer was avoided at 20 years for every 1.5 recurrences avoided during the first 10 years after radiotherapy". It should be noted that these analyses relate to a time when systemic treatments were not as advanced and widespread, which could to some extent impact their direct translation to the present day. However, the incorporation of radiotherapy into the multidisciplinary treatment of breast cancer offers a significant benefit even in those patients who achieve a pCR after PST [22].

Although its use before surgery is not commonly considered, the use of preoperative radiotherapy in localized breast cancer is far from new and has shown that this therapeutic alternative is feasible, well tolerated, and associated with a complete pathological response rate of 10–26% (Table 1) [23–25].

**Table 1.** Studies of preoperative radiotherapy in breast cancer.

| Author | *n* | Inclusion Criteria | RT | % pCR | Number of Local Recurrences | DFS (%) | OS (%) | Skin Complications | MFU (Months) |
|---|---|---|---|---|---|---|---|---|---|
| Semiglazov et al., 1994 [23] | 134 | IIB–IIIA | 60 Gy/2 Gy/30 fractions; SCF 40 Gy/2 Gy/20 fractions | 19.4% | NS | 5y DFS: 71.6 | 5y OS: 78.3 | G2: 8.9% | 53 |
| Calitchi et al., 2001 [24] | 75 | T2–T3 | 45 Gy/1.8 Gy/25 fractions whole breast, lower axillary nodes | 11% | 9 | 10y DFS: 47 | 10y OS: 55 | 6% poor cosmetic results | 120 (10 years) |
| Riet et al., 2017 [25] | 187 | T2–T4 | 45 Gy/2.5 Gy/18 fractions | 10% (26% in TN tumors) | 15 | 25y DFS: 30 | 25y OS: 30 | Post-operative G ≥ 2: 19% | 384 (32 years) |

MFU: median follow-up.

Other groups have published results that contribute both to reinforcing the safety and usefulness of preoperative radiotherapy and to facilitating the identification and selection of breast cancer patients who could benefit most. Deng et al. [26] published an analysis of 41,618 women with locally advanced breast cancer included in the National Cancer Data Base (NCDB) between 2010 and 2014. Although a small percentage of patients with LABC received preoperative radiation, the authors performed a propensity score matching analysis between patients who received preoperative or postoperative radiotherapy and observed similar outcomes with no differences relating to the timing of the radiotherapy. Due to the limited data, further subgroup analysis of the length of the radiotherapy and the patients' dose was not performed. Similarly, Zhang et al. [27] studied 411,279 women diagnosed with stage I–III infiltrating breast carcinoma treated between 1975 and 2016 and included in the Surveillance Epidemiolo Gy and End Results (SEER) database. With the limitations inherent in this type of analysis, the authors' conclusion is that preoperative radiotherapy does not offer survival advantages over postoperative radiotherapy, but it might have other advantages that are worthy of further exploration. Despite the differences between them, the unbalanced number of patients and the uneven follow-up period, final outcomes from both studies suggest that preoperative radiotherapy is feasible, with a low toxicity profile and with no disadvantages compared to adjuvant radiotherapy. Poleszczuk et al. [28] published their results based on an analysis of the large SEER database, showing that preoperative radiotherapy in breast cancer is safe and does not decrease overall survival in patients with localized tumors. The authors pointed out, moreover, that radiotherapy administered to a large tumor bulk activates robust antitumor immunity, a fact that would be absent when radiotherapy is administered after surgery, and that

this radio-induced immunity could contribute to eliminating not only the primary tumor but also microscopic foci present in the ipsilateral and contralateral breast, as well as diminishing the risk of distant micrometastasis, leading to an abscopal effect of preoperative radiotherapy. The authors also specified that in more aggressive tumors, immunity induced by preoperative irradiation would require potentiation with systemic agents, such as the concurrent administration of taxanes, to maximize effectiveness. Finally, Liu et al. [29], also based on data from SEER, proposed a nomogram for predicting cancer-specific survival in breast cancer patients undergoing preoperative radiotherapy.

## 2.2. Neoadjuvant Radiochemotherapy

Although frequent in other tumors where concurrent chemoradiotherapy is widely practiced (i.e., head and neck, esophagus, stomach, rectal, uterine cervix, or lung cancer), showing an increase not only in local control but also in survival rates, the combination has not been a widespread practice in breast cancer patients. In spite of the fact that surgery followed by chemotherapy then followed by radiation is considered to be the more conventional approach to breast cancer multidisciplinary treatment, the increasing use of neoadjuvant treatments has renewed the interest in exploring combined chemotherapy and radiation therapy in breast cancer, especially in the most aggressive and unfavorable molecular subtypes.

In Table 2 we summarize some of the studies of preoperative chemo-radiotherapy published to date. From 1994 to 2022, results from 24 different studies—14 prospective and 10 retrospective—including a total of 3460 breast cancer patients, were reported [23,30–52].

Most groups that have studied the preoperative administration of radiotherapy included in Table 2 used an irradiation schedule with an old fractionation of 1.8–2 Gy/day between 23 and 28 fractions. However, these fractionations are currently considered to be outdated and moderate hypofractionation at a dose of 4000–4250 c Gy in 15–16 fractions is the preferred schedule for women with breast cancer requiring postoperative radiotherapy on the breast/chest wall and regional nodal areas [30,31].

All but 2 used conventional 1.8–2 Gy/day up to 40–50.4 Gy over the breast and lymph node and 10 groups also considered a tumor bed boost, sequentially in 9 of them and simultaneously integrated into the study byCiérvide et al. [32]. Ho et al. [33] used a moderate hypofractionation schedule of 42.5 Gy in 16 fractions in 12 patients (40% of all patients treated) while Ciérvide et al. treated all their patients with a schedule of 40.5 Gy in 15 fractions.

**Table 2.** Studies of preoperative chemo-radiotherapy published to date.

| Author | n | Type of Study | Preoperative Radiotherapy | Preoperative Systemic Treatment | pCR (%) | Acute Toxicity | Postoperative Complications | DFS (%) | OS (%) | MFU (Months) |
|---|---|---|---|---|---|---|---|---|---|---|
| Semiglazov 1994 [23] | 137 | Prospective | 60 Gy (2 Gy) WBI, 40 Gy (2 Gy) RNI | TMF | pCR: 29% | G1-2: 6.5% | 20% | 5y: 81 | 5y: 86 | 53 |
| Skinner, 1997 [34] | 35 | Prospective | 50 Gy (2 Gy) WBI + RNI | Pre-RT: F → Concurrent: F | pCR: 17% | NR | NR | 2y: 83 | 2y: 90 | 22 |
| Touboul 1997 [35] | 97 | Prospective | 46 Gy (2 Gy) WBI + RNI | Sequential: CAF-V × 4 | pCR: 40% | NR | NR | 10y: 60 | 10y: 66 | 94 |
| Colleoni 1998 [36] | 29 | Prospective | 50 Gy (2 Gy) WBI + boost to tumor nodule (10 Gy) | Sequential: AC × 3 | pCR: 6% | NR | NR | NS | NS | NS |
| Skinner2000 [37] | 28 | Prospective | 45 Gy (1.8 Gy) WBI + RNI | Concurrent: TAX × 8 | pCR:26% | NR | 41% | NS | NS | NS |
| Aryus 2000 [38] | 56 | Prospective | 50 Gy (2 Gy) WBI/RNI + boost tumor | Sequential: CMF or EC | pCR: 43% | NR | NR | NS | NS | NS |
| Formenti 2003 [39] | 44 | Prospective | 45–46 Gy (1.8–2 Gy) WBI + RNI | Pre-RT: dTAX → Concurrent: dTAX | pCR 34% | G2: 45% G3: 7% | NR | 3y: 75.6 | 3y: 94 | 32 |
| Gerlach 2003 [40] | 134 | Retrospective | 50 Gy (2 Gy) WBI/RNI | Sequential: EC + CMF (n = 50) or EC | pCR: 42% | NENR | NR | NS | NS | 19 |
| Lerouge 2004 [41] | 120 | Prospective | 46 Gy (2 Gy) WBI + RNI | Sequential CAF-V × 4 or C-THP-F-Vd | pCR: 35% | NR | NR | 10y: 61 | 10y: 66.5 | 140 |
| Chakravarthy 2006 [42] | 38 | Prospective | 45 Gy (1.8 Gy) WBI + RNI | Pre-RT: TAX → Concurrent: TAX | pCR: 34% | G3: 2.6% G4: 2.6% | 10% | NS | NS | 23 |
| Bollet 2006 [43] 2012 [44] | 60 | Prospective | 50 Gy (2 Gy) WBI ± boost 10 Gy; 46 Gy (2 Gy) RNI | Concurrent: FVb × 6 | pCR: 27% | G2: 19% G3: 14% | 12% | 83 | 88 | 84 |
| Gaui 2007 [45] | 28 | Retrospective | 50 Gy (2 Gy) WBI + RNI | Concurrent: Capecitabine | pCR: 4.3% | G1: 35% G2: 11% | 4% | NS | NS | NS |

**Table 2.** *Cont.*

| Author | n | Type of Study | Preoperative Radiotherapy | Preoperative Systemic Treatment | pCR (%) | Acute Toxicity | Postoperative Complications | DFS (%) | OS (%) | MFU (Months) |
|---|---|---|---|---|---|---|---|---|---|---|
| Shanta 2008 [46] | 1117 | Retrospective | 40 Gy (2 Gy) WBI + RNI | Concurrent: CMF/ECF/FAC | pCR: 45.1% | NR | 5.8% | 52.6 | 63.9 | NS |
| Alvarado-Miranda 2009 [47] | 112 | Retrospective | 50 Gy (2 Gy) WBI + RNI + boost 10 Gy | Concurrent: McF or cDDP-GMZ | pCR: 29.5% | G3: 22.4% | 17% | 76.9 | 84.2 | 43 |
| Adams 2010 [48] | 105 | Pooled analysis from three prospective trials, including Bollet 2006 and Formenti2003 | 45 Gy (1.8 Gy) WBI + RNI ± boost 14 Gy | Concurrent: TAX +/− Trastuzumab | pCR: 23% | NR | NR | 61.4 | 71.6 | 60 |
| Monrigal 2011 [49] | 210 | Retrospective | 50 Gy (2 Gy) WBI + RNI + boost 10 Gy | Concurrent: Anthracyclin-based CT ± TAX ± Trastuzumab | pCR: 35.2% | NR | 26% | 5y:75.6 | 5y: 86.7 | 120 |
| Daveau 2011 [50] | 165 | Retrospective | 45 Gy (1.8 Gy) WBI + RNI + boost to tumor nodule (10–15 Gy) | Sequential: CAF or AdTAX × 6 cycles | pCR: 41% | NR | NR | 65 | 91 | NS |
| Ho 2012 [33] | 120 | Retrospective | 50 Gy (2 Gy) (60%) or 42.5 Gy (2.67 Gy) (40%)WBI/RNI; Boost (median dose 6.25 Gy) | Sequential: AC (n = 1); CMF (n s = 6); FEC (n = 4); A-TAX (n = 15); | pCR: NE | NR | 37% | 5y: 65 | 5y: 68 | 42 |
| Zinzindohoue 2016 [51] | 83 | Prospective | 50 Gy (2 Gy) WBI + RNI | Sequential: A-TAX | pCR: 36% | NR | 6% | 2y: 68 | NS | 24 |
| Brackstone 2017 [52] | 32 | Prospective | 45 Gy (1.8 Gy) WBI + RNI ± boost 5.4 Gy | Pre-RT: FEC → Concurrent: dTAX | pCR: 22.6% | G3: 25% | 3% | 3y: 81 | 3y: 89 | 36 |
| Pazos 2017 [53] | 22 | Retrospective | 50.4 Gy (1.8 Gy) WBI + RNI | Sequential: EC → TAX | pCR: 5% | NR | 25% | 30 months: 18 | 30 months: 18 | 30 |

**Table 2.** *Cont.*

| Author | n | Type of Study | Preoperative Radiotherapy | Preoperative Systemic Treatment | pCR (%) | Acute Toxicity | Postoperative Complications | DFS (%) | OS (%) | MFU (Months) |
|---|---|---|---|---|---|---|---|---|---|---|
| Haussmann 2022 [54] | 356 | Retrospective | 50 Gy (2 Gy) WBI + RNI + boost 10 Gy | Sequential: EC/CMF/AC/Mitoxantrone 61% or Concurrent: 36% or No CHT: 3% | pCR: 31.1% | NR | NR | NS | 10y: 69.7; 20y: 53.1 | 240 |
| Ciérvide 2022 [32] | 58 | Prospective | 40.5 Gy (2.7 Gy) WBI + RNI + SIB 54 Gy (3.6 Gy) | Concurrent: Pertuzumab-Trastuzumab-TAX → AC in HER2+ Concurrent: CBDCA-TAX → AC in TNBC | TN: 71% HER2+ 53% HR+: 48% HR−: 64% | G1: 78% G2: 14% G3: 5% | 16% | 100 | 96.5 | 24 |

MFU: median follow-up; TMF: thiotepa-methrotexate-5Fluoracil; CAF-V: cyclophosphamide-adriamycin-5Fluoracil-vincristine; AC: adriamycin-cyclophosphamide; F: 5Fluoracil; TAX: paclitaxel; C-THP-F-Vd: cyclophosphamide-tepirubicin-5fluoracil-vindesine; CMF: cyclophosphamide-methrotexate-5fluoracil; Mc: mitomycin-C; cDDP: cisplatin; GMZ: gemcitabine; EC: epirubicin-cyclophosphamide; A-dTAX: adriamycin-docetaxel; FVb: 5fluoracil-vinorelbine; AI: aromatase inhibitor; A-TAX: adriamycin-paclitaxel; NR: not reported; NS: not specified.

Although there are no specific studies analyzing fractionation in preoperative radiotherapy in breast cancer, the clinical results that support the use of moderate hypofractionation schedules after surgery also allow their application in the preoperative setting. The Ciérvide et al. [32] group recently published the results of a phase II study of preoperative concurrent radiochemotherapy in patients with TN and HER2+ breast cancer using a hypofractionated schedule of 40.5 Gy in 15 fractions over breast and lymph node chains with a simultaneous integrated boost of up to 56 Gy in 15 fractions over macroscopically affected areas in breast and lymph node regions. The authors report an acceptable toxicity profile with only the presence of mild (76% G1) or moderate (12% G2) radio-induced skin toxicity and a rate of postoperative complications similar to those described in other studies with more conventional fractionation. Vincent et al. [55] have published the results of the POP-ART randomized trial comparing preoperative radiotherapy followed by chemotherapy and surgery with neoadjuvant chemotherapy followed by surgery and postoperative radiotherapy using in both cases an ultra-hypofractionation schedule of 28.5 Gy with SIB up to 31 Gy administered in five fractions every other day in 20 patients diagnosed with stage I–II breast cancer. The primary objectives of the study comprised an analysis of the feasibility and safety of preoperative treatment and differences regarding total treatment time (OTT) between the two modalities. In the 19 patients analyzed, the authors found no differences with respect to the administration of chemotherapy, radiotherapy, or surgery in the outcomes and complications. However, preoperative administration of radiotherapy was associated with shorter OTT.

Different chemotherapy schedules have been used, both sequentially prior to radiotherapy (11 studies) and concurrently with preoperative radiotherapy (13 studies). Only a few studies have used targeted drugs and most of them include taxanes in their schemes.

## 3. Clinical Outcomes

Despite all the differences between the studies included, the overall objective response rate ranged from 64% to 93% and the pCR rate ranged from 5% to 71% (median 53%). One of the most important limitations of these studies is the lack of information regarding the toxicity profile. Considering the available information, five studies reported acute cutaneous toxicity equal or greater than grade 3 in 5–22% of patients (median 7%) and 12 studies recorded the occurrence of post-surgical complications ranging from 3 to 41% of treated patients (median 14%).

One of the arguments most widely used to call into question the usefulness of preoperative radiotherapy in breast cancer is the potential complications that it could have for breast surgery, especially when a mastectomy is required, and the impact that altering the sequence of treatments might have on subsequent reconstruction. Delayed breast reconstruction is generally preferred in order to decrease the risk of unfavorable surgical outcomes, especially with regard to longer-term aesthetic results [56]. However, existing evidence points in a different direction. Preoperative radiotherapy allows for a single-stage surgical procedure, significantly shortening overall breast cancer treatment time, improving patient breast satisfaction, and utilizing limited health resources more efficiently. Importantly for the patient, at no time is there a breast tissue deficit as the breast reconstruction is completed immediately. From an oncological point of view, the risk of any delay in the start of post mastectomy radiotherapy (PMRT) from prolonged surgical recovery or complications is avoided; dosimetric coverage of targets is more achievable when there are no reconstructive materials in place; and there is a potential increase in the likelihood of pathological downstaging and achieving a R0 resection. Although longer follow-up and larger prospective trials are required to review late complication rates and oncological outcomes, evidence is emerging. Baltodano et al. [57] have analyzed the results observed in 77,902 patients included in the American College of Surgeons National Surgical Quality Improvement Program (ACS-NSQIP) in 2005–2011 treated by mastectomy alone (78.4%) or mastectomy with immediate reconstruction (21.6%). The authors reviewed the frequency of surgical or reconstructive complications. Of the patients in both groups, 0.4% had pre-

operative radiotherapy. The results obtained did not show the existence of complications in relation to the administration of radiotherapy before surgery in the performance of a mastectomy or with respect to immediate reconstruction, concluding that preoperative radiotherapy is a safe and feasible procedure when a mastectomy with or without immediate reconstruction is planned. Singh et al. [58] performed a systematic review of 18 retrospective and prospective studies of preoperative radiation therapy and immediate breast reconstruction including a total of 1047 patients, which showed the safety of this treatment, both technically and oncologically. Thiruchelvam et al. [59] have presented the results of the PRADA study which analyzed the incidence of complications in the form of suture dehiscence greater than 1cm in 33 consecutive patients treated with PST and preoperative radiotherapy with moderate hypofractionation followed by mastectomy 2–6 weeks after, with deep inferior epigastric perforator (DIEP) flap reconstruction. Four weeks after surgery, four patients (12.1%) suffered suture dehiscence greater than 1cm. The authors' conclusion is that reconstruction using DIEP after PST and preoperative radiotherapy is feasible and safe, with surgical wound dehiscence rates similar to those of post-mastectomy radiotherapy. At the recent ASTRO 2022 meeting, Admojo et al. presented the results of a retrospective analysis conducted on 155 patients treated with radiotherapy prior to mastectomy and DIEP reconstruction compared to the results observed in 31 women who underwent mastectomy and DIEP reconstruction followed by postoperative radiotherapy. The objectives of the study were to analyze the complication rates in the form of flap contracture, fat necrosis, and cosmetic outcome in both groups. The researchers observed a significantly higher incidence of flap contracture (41.9% vs. 1.9%, $p < 0.001$) and fat necrosis (19.4% vs. 12.9%, $p < 0.001$) when radiotherapy was administered postoperatively and a good/excellent cosmetic outcome in 96.1% of women when radiotherapy was administered preoperatively vs. 80.6% when administered postoperatively ($p < 0.001$).

Existing evidence supports the feasibility of preoperative administration in breast cancer and its integration into accepted PST schemes. The rates of pCR are encouraging, especially in molecular subtypes such as TN or HER2-enriched, considered to be more aggressive. Nevertheless, and despite this evidence, concerns have been raised about the safety of simultaneous administration of chemotherapy and radiotherapy in breast cancer, and more specifically regarding the use of potential cardiotoxic drugs such as trastuzumab, pertuzumab, or taxanes.

The preoperative and postoperative combination of radiotherapy and taxanes is not only effective and safe but is also linked to better survival, especially in the setting of patients suffering a loss of hormonal receptor expression and Her-2 enriched phenotypes. In addition, the synchronic delivery of cardiotoxic agents, such as trastuzumab, pertuzumab, or a combination of both, plus locoregional radiotherapy, has been shown to be safe and well tolerated without increasing adverse cardiac effects, not only with conventional but also with hypofractionated schemes, even when the internal mammary chain has to be irradiated [60–65]. Furthermore, new high conformal radiation techniques and systems used to track body motion (SGRT, Surface Guided Radiation Therapy) help minimize the dose in nearby healthy organs at risk and improve treatment precision and safety.

## 4. Neoadjuvant Accelerated Partial Breast Irradiation (APBI)

Breast-conserving therapy consisting of breast-conserving surgery (BSC) followed by whole breast irradiation (WBI) is the standard treatment for early-stage disease. Since the risk of local recurrence is low, and most local recurrences are located within the vicinity of the surgical bed, accelerated partial breast irradiation (APBI) has been considered as an alternative to WBI. As regards target volume, definition is more precise before surgery and a substantial reduction in treatment volumes can be achieved with preoperative APBI, when compared to a post-operative approach. Preoperative APBI has been researched in women with early-stage and low-risk breast cancer due to its potential to minimize RT treatment duration and toxicity. A few studies of preoperative APBI have been published (Table 3). Bondiau et al. [66] were among the first authors to show that SBRT can be safely

combined with NACT and delivered promising results in terms of the pCR rate (36%) and breast conservation percentages (92%). Horton et al. [67] investigated single-dose radiation therapy (15, 18, or 21 Gy) in a small study with 32 patients. After a median follow-up of 23 months, no recurrences were detected and only grade 1 to 2 toxicities and good to excellent cosmetic outcomes in all patients were reported. Van der Leij et al. [68] recorded good outcomes in terms of limited fibrosis and excellent cosmetic results with excellent results at 3 years of follow-up. Nichols et al. [69] also reported limited toxicity and good to excellent cosmetic outcomes after a median follow-up of 3.6 years in 27 patients treated with preoperative APBI (38.5 Gy in 3.85 Gy fractions delivered twice daily), followed by breast-conserving surgery after 21 days. Guidolin et al. [70] did not identify any significant toxicity and demonstrated excellent cosmetic and quality-of-life outcomes with a single APBI dose. Weinfurtneret al. [71] showed how MRI could be a precise tool to evaluate and predict breast cancer response after pre-operative SABR treatment. Bosma et al. [72] concluded, based on their results, that preoperative APBI is a feasible method with a low postoperative complication rate, limited fibrosis, and a good to excellent cosmetic outcome. The low complete pathological response rate in the studies published is remarkable. Although survival outcomes are good, in all the studies there was a residual tumor in the surgical specimen. One of the possible causes is the short interval of time between radiotherapy and surgery, suggesting the delayed anti-tumor effect of radiotherapy may not yet have been expressed.

In summary, the feasibility and the efficacy of a pre-operative radioablative approach on early breast cancer patients have been explored in a few clinical studies, using different techniques, dose/fraction, number of fractions, total dose, and irradiated volumes. Up-to-date, preliminary reports seem to show low toxicity and an acceptable rate of pathological response.

**Table 3.** Published preoperative accelerated partial breast irradiation studies.

| Author/Year of Publication | Inclusion Criteria | *n* | Radiotherapy | Chemotherapy | Time Interval to Surgery | pCR | Outcomes | Late Toxicity | Follow Up (Months) |
|---|---|---|---|---|---|---|---|---|---|
| Bondiau et al., 2013 [66] | Unifocal Not suitable for BCSHer-2- | 26 | SBRT (19.5–31.5 Gy/3fx) Dose escalation level (19.5 Gy, 22.5 Gy, 25.5 Gy, 28.5 Gy or 31.5 Gy) | Neoadjuvant chemotherapy: TAX × 3 → FEC × 3 | 4–8 weeks after last chemo cycle | 36% | 92% BCS 96% objective response rate (ORR) | 0 | 30 |
| Horton et al., (DUKE study) 2015 [67] | Age > 55y T1 DCIS G1-2 < 2 cm cN0ER, PR+, Her-2+ | 32 | IMRT (15–21 Gy/1fx) | No chemotherapy | Within 10 days after RT | NR | 0% recurrences | 13 G2 2 G3 PRCO were good/excellent | 23 |
| Van der Leij et al., 2015 [68] | Age > 60y Invasive, unifocal, non-lobular, T < 3 cm Negative SLNB | 70 | 3DRT or IMRT or VMAT40 Gy/10fx | No chemotherapy | 6 weeks after RT | NR | 2 ipsilateral breast tumor recurrence | 11% G2 induration at 12 months 2% G2 fibrosis at 24 months | 23 |
| Nichols et al., 2017 [69] | Invasive, unifocal T < 3 cm cN0 | 27 | 3DRT 38.5 Gy/10fx | No chemotherapy | >21 days after RT | 15% | 88.9% ORR 70.4% of Ki 67 reduction after RT | PRCO fair (17%) and poor (5%) at 1y | 43.2 |
| Guidolin et al., (SIGNAL study) 2019 [70] | Ductal, unifocal, postmenopausal, T < 3 cm, ER+, cN0, invasive, tumor at least 2 cm away from skin and chest wall | 27 | 21 Gy/1fx | No chemotherapy | 1 week after RT | NR | 100% alive and free from recurrence | 1y toxicity, PRCO, and HRQoL were not significantly different from baseline | 16.2 |
| Bosma (PAPBI) et al., 2021 [72] | >60y, invasive, unifocal, non-lobular pNO (determined by SLNB) | 133 | 40 Gy/10fx in 2 weeks (2010–2013) 30 Gy/5fx in 1 week (after 2013) $^{18}$FDG–PET pre- and post-RT | No chemotherapy | 6 weeks | 23% | 3 local recurrences 1 ipsilateral breast recurrence | 5y excellent to good cosmesis 90% | 60 months |
| Weinfurtner (SABR study) et al., 2022 [71] | >50y, cT1-2, ER/PR + HER2− | 19 | SBRTbaseline breast MRI, and presurgical MRI 28.5 Gy/3fx | No chemotherapy | 5–6 weeks | 0% | NR | NR | NR |

## 5. New Perspectives of Neoadjuvant Radiotherapy: Ongoing Trials

Novel trials involving preoperative radiotherapy have been initiated. Table 4 is a compilation of 22 studies that aim to analyze the usefulness of preoperative radiotherapy, in its different variants, in breast cancer: six studies use WB $\pm$ RNI, 11 trials APBI, and 5 studies the administration of radiotherapy as an anticipated boost prior to surgery and subsequent adjuvant radiotherapy. Most of them are phase I/II trials although there are four randomized phase III studies, with WBI (NCT05512286, NCT04261244), APBI (NCT03875573), and an anticipated boost (NCT03804944).

Of particular interest is the fact that three studies (NCT03366844, NCT03875573, and NCT03804944) plan to analyze the combination of radiotherapy and immunotherapy. Advances in modern immunotherapy are contributing to definitive changes in the landscape of oncolo Gy. The development of drugs specifically aimed at boosting the immune response against cancer represents a paradigm shift. Combining radiation therapy and immunotherapy is not a novel or unfounded hypothesis. The ability of radiotherapy to modulate different stages of the immune response against cancer, from the generation and release of tumor antigens to the possibility of inducing cell death mediated by T lymphocytes, may make the combination of both strategies a promising alternative for these women.

The clinical objectives of ongoing trials vary, as do the primary endpoints of the studies, which include assessing feasibility, pathologic complete response rate, identifying the maximum tolerated dose, analyzing acute toxicity and surgical complications, and cosmetic outcome. Two studies (NCT03366844 and NCT03359954) also seek to evaluate changes in tumor-infiltrating lymphocytes (TIL). Differences exist regarding the interval from radiation to surgery, between 1 to 52 weeks; hence, it is expected that outcomes in terms of pathological response will be different. Three studies have already been completed, nine are recruiting patients, eight are not recruiting, and the status of one is unknown. The results of these and other upcoming trials will contribute to enhancing knowledge of breast cancer radiation biolo Gy and redefine the role of preoperative radiotherapy in different scenarios.

**Table 4.** Ongoing preoperative RT trials.

| NCT Number | Locations | Type of Study | Patients | N Estimated | Description | Time to Surgery | Objectives | Status |
|---|---|---|---|---|---|---|---|---|
| WBI | | | | | | | | |
| NCT05512286 (CAPPELLA) | Guangxi, China | Randomized phase III | cT0-3, T4b and cN0-3a | 80 | Radiotherapy followed by mastectomy and DIEP flap reconstruction vs. radiotherapy after mastectomy and DIEP flap reconstruction | 2–6 weeks | Patient satisfaction | Not yet recruiting |
| NCT05412225 | New York, NY, USA | Phase II | cT4 cN0-3 | 60 | T4 M0 breast cancer patients with complete or partial response to standard neoadjuvant chemotherapy and immediate autologous reconstruction | 2–6 weeks | Wound complications | Recruiting |
| NCT05274594 | Istanbul, Turkey | Phase II | cT1-3 cN + | 37 | WBI + RNI: 42.5 Gy/16fx or 50 Gy/25fx or 50.4 Gy/28fx | 6 weeks | Pathologic complete response | Completed |
| NCT04261244 (NEORAD) | Duesseldorf, Germany | Randomized phase III | cT2-T4 (non-inflammatory) cT1, if G3, * triple negative, Her2 positive, or cN+ | 1826 | Preoperative radiotherapy in breast cancer after neoadjuvant chemotherapy vs. postoperative radiotherapy after neoadjuvant chemotherapy | 3–8 weeks | DFS | Not yet recruiting |
| NCT03624478 | Scottsdale, Jacksonville, Rochester, NY, USA | Phase II | cT0-T2 cN0 | 25 | Preoperative ultra-hypofractionated WBI | 4–16 weeks | Pathologic complete response | Active, not recruiting |
| NCT02858934 | Brussels, Dendermonde, Belgium | Phase II | cT1-2N0M0 | 24 | WBI 25 Gy in 5 daily fractions of 5 Gy, SIB 30 Gy in 5 daily fractions of 6 Gy | 1 week | Duration of surgical procedure, blood loss, wound complications | Completed |
| APBI | | | | | | | | |
| NCT01014715 (GCC 0919) | Baltimore, MD, USA | Phase II | Tc1-2 cN0 | 32 | Preoperative radiation followed by lumpectomy | 3 weeks | Reproducibility of delivering preoperative APBI in Stage I and Stage IIA breast cancers | Completed |

**Table 4.** *Cont.*

| NCT Number | Locations | Type of Study | Patients | N Estimated | Description | Time to Surgery | Objectives | Status |
|---|---|---|---|---|---|---|---|---|
| NCT05464667 | Pittsburgh, PA, USA | Phase I/II | cTis-1 cN0 Luminal | 24 | Dose Escalation: 5 Cohorts—30 Gy in 5 fractions (baseline treatment with 0 boost dose to GTV), 35, 40, 45, 50 Gy in 5 fractions (Part 1) Dose Expansion: Maximum Tolerated Dose determined during dose escalation (Part 2) | NR | Maximum tolerated dose | Not yet recruiting |
| NCT02316561 (ABLATIVE-1) | Amsterdam, Netherlands | Phase II | cT1 cN0 (<50y) cT1-2 (≤3 cm) cN0 (>70y) | 25 | Single dose of 20 Gy/15 Gy on the gross tumor volume and clinical tumor volume respectively | 24 weeks | Pathologic complete response | Completed |
| NCT05350722 (ABLATIVE-2) | Amsterdam, The Netherlands | Phase II | cTis-1 cN0 Luminal | 100 | Single dose of 20 Gy/15 Gy on the gross tumor volume and clinical tumor volume respectively | 24 weeks | Pathologic complete response | Recruiting |
| NCT05217966 (SPtedORT-DNS) | Montreal, QC, Canada | Phase II | cTis-1 cN0 Luminal | 80 | Single Pre-Operative Radiation Therapy | 52 weeks | Pathologic complete response | Recruiting |
| NCT04679454 (CRYSTAL) | Milan, Italy | Phase I/II | cT1-T2 (up to 2.5 cm) cN0 | 79 | Phase I: 3 dose levels:18 Gy, 21 Gy and 24 Gy in single fraction phase II: clinical evaluation | 4–8 weeks | Phase I: maximum tolerated dose Phase II: pathologic complete response of selected dose | Recruiting |
| NCT04360330 (SABER) | Miami, FL, USA | Phase I | cT1 cN0 Luminal | 18 | Preoperative SABR Phase I study testing up to 4 dose levels: 35 Gy (5 fractions of 7 Gy); 40 Gy (5 fractions of 8 Gy); 45 Gy (5 fractions of 9 Gy); 50 Gy (5 fractions of 10 Gy) | 4–6 weeks | Recommended dose for a phase II | Recruiting |

**Table 4.** *Cont.*

| NCT Number | Locations | Type of Study | Patients | N Estimated | Description | Time to Surgery | Objectives | Status |
|---|---|---|---|---|---|---|---|---|
| NCT03875573 (NEO-CHECK-RAY) | Brussels, Belgium | Randomized phase II | cT2 zN0 or cT1 cN1-3 Luminal B HER2- | 147 | Preoperative 3 × 8 Gy with chemotherapy ± durvalumab ± oleclumab | 2–6 weeks | Immune related or radiation therapy related toxicity | Recruiting |
| NCT02728076 | Milwaukee, WI, USA | Phase II | Clinically stage I-II | 40 | Preoperative MRI-based radiation followed by lumpectomy | 5–8 weeks | Postoperative complications | Active, not recruiting |
| NCT02482376 | Durham, CN, USA | Phase II | cTis-1 cN0 Luminal | 68 | Single fraction of 21 Gy of stereotactic radiotherapy before proceeding to surgery. | 2–4 weeks | Physician reported rates of good/excellent cosmesis | Active, not recruiting |
| NCT02065960 (ARTEMIS) | Hamilton, ON, Canada | Phase II | cT1 cN0 Luminal | 32 | SABR to a dose of 40 Gy in 5 fractions delivered every other day over a period of 10–12 days, followed by breast conserving surgery | 8–12 weeks | Feasibility | Unknown status |
| ANTICIPATED BOOST | | | | | | | | |
| NCT05603078 (BIRKIN) | Beijing, China | Phase II | cT1-4 cN0 | 102 | Preoperative MRI-guided tumor-bed boost and post-operative ultra-hypofractionated radiotherapy (26 Gy/5.2 Gy/5) | 4 weeks | Primary endpoint: acute toxicities; secondary endpoints: oncologic outcomes, surgical complications within 30 days, late toxicities, patients' quality of life and cosmetic outcomes. | Recruiting |
| NCT04871516 | New Brunswick, NJ, USA | Phase II | Clinical stage 0-IIIC | 55 | Anticipatedboost in 4 fractions | 1–3 weeks | Wound complications | Recruiting |
| NCT03366844 | Los Angeles, CA, USA | Phase I/II | T2-4c cN0-3 any subtype | 60 | Anticipated boost 3 × 8 Gy + pembrolizumab | 6 weeks | Feasibility and changes in tumor-infiltrating lymphocytes (TIL) | Active, not recruiting |

**Table 4.** *Cont.*

| NCT Number | Locations | Type of Study | Patients | N Estimated | Description | Time to Surgery | Objectives | Status |
|---|---|---|---|---|---|---|---|---|
| NCT03804944 | New York (NY), Pittsburh (PA), Houston (TX), USA | Randomized Phase III | Clinical stage II-III ER + HER2- | 100 | Anticipated boost $3 \times 8$ Gy + letrozole $\pm$ pembrolizumab or Ftl-3 ligand or pembrolizumab + Ftl-3 ligand | 14 weeks | Feasibility, clinical response, pathologic response | Active, not recruiting |
| NCT03359954 (PRECISE) | Houston, TX, USA | Phase II | cT1-4 cN0-3 Luminal | 25 | Anticipated boost | 1 week | Changes in tumor-infiltrating lymphocytes (TIL) | Active, not recruiting |

WBI: whole breast irradiation; RNI: regional node irradiation; APBI: accelerated partial breast irradiation; MRI: magnetic resonance image; SABR:stereotactic ablative breast radiotherapy; SIB: simultaneous boost integrated; DFS: disease-free survival; Flt-3L: FMS-related tyrosine kinase 3 ligand.

## 6. Conclusions

Breast cancer treatment paradigms are constantly evolving, abandoning rigid rules considered almost immutable until very recently. The temporal sequence of the different treatments involved is changing, and it is not possible to advocate a specific regimen as the standard, but these must be adapted to the conditions of each breast cancer patient. Constant improvements in systemic treatments, through the design of regimens that are increasingly personalized and tailoredto the particularities of the different breast cancers, are modifying the old concept of chemotherapy. Similarly, the evolution of radiotherapy for breast cancer, with the generalization of radiation schedules with moderate hypofractionation or ultra-hypofractionation and supported by modern technological advances that allow treatments to be carried out with high precision and safety, favors the change in perspective in the multidisciplinary management of breast cancer.

Thus, the administration of systemic treatment prior to surgery is now standard practice and, likewise, preoperative radiotherapy in selected cases is worthy of consideration because neoadjuvant radio(chemo)therapy in selected patients could offer some clinical advantages:

- First, a preoperative approach allows for the definition of more precise targets than in the postoperative setting, especially regarding the boost volume that could be blurred by the potential existence of fibrosis and/or seroma and the presence or not of surgical clips that may be changeable. It would decrease the risk of geographic loss associated with the postoperative delimitation of the volumes of interest for radiotherapy, especially regarding the increasing interest in the use of oncoplastic surgical techniques and the challenge in localizing the tumor bed due to tissue rearrangement, hindering the safe administration of this boost.
- Second, preoperative irradiation facilitates immediate reconstruction in those patients undergoing a mastectomy, reducing delay intervals, and probably contributing to a better cosmetic result by avoiding flap irradiation and the associated risk of shrinkage and fibrosis.
- Third, radiotherapy before surgery could favor the use of skin-sparing mastectomy techniques for reconstruction by minimizing the risk of postmastectomy residual disease as this tissue would have already been irradiated.
- Fourth, evidence supports the hypothesis that tumors develop multiple immune evasion mechanisms as they progress, and some cancers are inherently better at "hiding" than others. It has been suggested, moreover, that radiotherapy applied to a large tumor bulk activates robust antitumor immunity, a fact that would be absent when radiotherapy is administered after surgery, and that this radio-induced immunity could contribute to eliminating not only the primary tumor but also microscopic foci present in the ipsilateral and contralateral breast as well as diminishing the risk of distant micrometastasis, leading to an abscopal effect of preoperative radiotherapy.
- Finally, simultaneous administration of radiotherapy and chemotherapy may also impact overall treatment time, by reducing the number of hospital visits, and contribute to improving patient satisfaction and therapeutic adherence while facilitating a reduction in the total cost of treatment.

Switching perspective in the multidisciplinary management of breast cancer will include a search for the personalization of different treatments, new combinations, and different ways of integrating them that will enable advances to be made in tailoring the approach to each individual woman with breast cancer.

**Author Contributions:** R.C. and A.M. contributed equally to all aspects of this paper: its conceptualization, methodolo Gy, and research as well asto the writing, reviewing, and editing of the manuscript. All authors have read and agreed to the published version of the manuscript.

**Funding:** This research received no external funding.

**Institutional Review Board Statement:** Not applicable.

**Informed Consent Statement:** Not applicable.

**Acknowledgments:** To Alyson Kim Turner, for reviewing, revising, and editing this manuscript, including English language grammar and syntax correction.

**Conflicts of Interest:** The authors declare no conflict of interest.

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
