# Peer review of "Preoperative Radio(Chemo)Therapy in Breast Cancer: Time to Switch the Perspective?"

_curroncol, doi:10.3390/curroncol29120768_

Round 1
Reviewer 1 Report
Thank you for the opportunity for review the manuscript of Montero and Ciervide on 'Preoperative radio(chemo)therapy in breast cancer: time to switch the perspective'
This is a generally a well written review of the topic of radio-chemotherapy in breast cancer. The literature review is comprehensive and up to date and the interpretation of the data generally well balanced.I think some mention should be made on progress in the identification of molecular markers of breast cancer radiosensitivity since identification of likely response or not from preoperative radiotherapy would be very important in selecting patients for preoperative radiotherapy before breast conserving surgery or mastectomy.
I have the following specific points.
There are several points in the manuscript where there should be gaps between phrases eg line 9 of the manuscript ('inperspective')
Title: I think it might be better to pose the title as a question rather than has a statement since there is insufficient level 1 evidence to make preoperative radiotherapy a standard of care for any group of patients at present.
Abstract: line 3 typo? a (r) evolution
Introduction
para 3, line 3 suggest 'altering' rather than 'alternating'
Neoadjuvant systemic treatment
para 2 last line reference(s) needed to survival gains
Neoadjuvant radiotherapy
para 1, line 8 add reference 13 at end of line
line 11, add reference 14 at end of line
Last para line 3 add reference to anti-tumor immunity
Neoadjuvant radiochemotherapy
para 1 line 4 clarification: ' concurrent chemoradiotherapy' rather than combination?
Clinical outcomes
Perhaps subheadings for breast conserving surgery and mastectomy
para 2 line 7 perhaps:' points in a different direction'?
Neoadjuvant accelerated partial breast irradiation (ABPI)
last line above Table 3
suggest altering to 'between radiotherapy and surgery, the delayed anti-tumor effect of radiotherapy may not yet have been expressed'
New Perspectives ? On Neoadjuvant Radiotherapy: ongoing trials
para 2 line 3 English 'to change definitively'
para 3 line 1 English 'as do' rather than 'and the'
line 5 'interval' rather than 'time lapse'
line 8 'one' instead of '1'
Conclusions
para 1, line 2 'suggest 'sequence' instead of 'organization'
Author Response
We thank the reviewer of Current Oncology for taking the time and effort to assess our manuscript.
We have thoroughly revised the manuscript according to the reviewer's suggestions and have introduced the pertinent changes:
- We have corrected the typographical errors present
- We have added the missing bibliographic references, rightly pointed out by the reviewer.
- Finally, we agree with the reviewer about the modification of the title, adding a question mark, something that we considered in the first versions of the manuscript although it did not appear in the version sent for review.
Reviewer 2 Report
I am grateful for the opportunity to review manuscript ID: curroncol-2059140 entitled „Preoperative Radio(chemo)therapy In Breast Cancer: Time To Switch The Prespective”.
The authors present a well structured, straightforward narrative review on the topic. The figures/tables are appropriate. I would recommend its publication after the authors address the comments below.
Please perform a language editing and eliminate some syntax errors in the manuscript.
Please consider elaborate the chirurgical aspects, e.g. complication risk due to preoperative irradiation, to provide a 360° view and narrative to the readers.
Some minor suggestions and questions:
Introduction, 1st paragraph: European Union consists of 27 member states. (https://european-union.europa.eu/principles-countries-history/key-facts-and-figures/structure_en) You mean maybe “40 European states” instead of “40 countries of the EU“?
Neoadjuvant systemic treatment, last paragraph: the cited trial was KeyNote-522 (not KN-22), and only one of the five main trials on Immuntherapy of TNBC (KN-522, Impassion 031, NeoTRIP, GeparNUEVO, I-SPY-2). You could maybe consider mention the latter, too.
Table 2: The cohorts reported by Roth 2010 and Matuschek 2012 seem to be the same. Four aspects have been reported yet by the working group:
- Roth 2010 (PMID: 20495968 DOI: 10.1007/s00066-010-2143-0) – pCR-rate, favourable outcome for cT2
- Matuschek 2012 (PMID: 22878547 DOI: 10.1007/s00066-012-0162-8) – long term outcome, predicitve factors
- Matuschek 2019 (PMID: 31101954 DOI: 10.1007/s00066-019-01473-2) – long term cosmesis
- Haussmann 2020 (PMID: 31919547 DOI: 10.1007/s00066-019-01557-z) – long term QoL
On Page 10, 2nd paragraph you mention 4 trials on immune therapy but only 3 of them are referenced.
Author Response
We thank the reviewer of Current Oncology for taking the time and effort to assess our manuscript.
We have thoroughly revised the manuscript according to the reviewer's suggestions and have introduced the pertinent changes:
- We have revised and corrected the syntax and typographical mistakes present in the manuscript.
- We have corrected the initial mention of European countries, avoiding the error of confusing it with the EU.
- We have corrected and added the references to the use of immunomodulatory drugs for TNBC, mentioning those studies suggested by the reviewer.
- We have revised Table 2 and adjusted its content to the latest versions of the published papers, avoiding redundancies, specifically with respect to the mentions of Roth et al., Matuschek at al. and Haussmann et al.
- Finally, we have corrected the existing error regarding number of ongoing trials with immunotherapy and preoperative radiotherapy.